# The Deep Wind Method: Physics-Informed Wind Field Reconstruction with Mass Consistency

Daniel Alejandro Cervantes Cabrera<sup>1</sup> and Miguel Angel Moreles Vázquez<sup>2</sup>

<sup>1</sup>INFOTEC, Centro de Investigación e Innovación en Tecnologías de la Información y Comunicación, México.

<sup>2</sup>CIMAT A.C., Centro de Investigación en Matemáticas, Jalisco S/N, Valenciana, 36240 Guanajuato, Gto., México.

**Correspondence:** Daniel Alejandro Cervantes Cabrera (daniel.cervantes@infotec.mx)

**Abstract.** We present the *Deep Wind* methodology, a physics-informed neural network (PINN) formulation for reconstructing three-dimensional wind fields from incomplete and noisy data. The approach embeds mass conservation and boundary conditions directly into the loss function, enabling physically consistent and stable reconstructions without mesh-based discretization. A series of synthetic benchmarks and real observations from Super Typhoon Kong-Rey (2024) demonstrate the robustness of the method compared to classical variational approaches. We show that Deep Wind consistently maintains stability and accuracy under sparse, irregular, or noisy observations. Overall, the results suggest that physics-informed deep learning is a promising framework for wind field recovery and data assimilation, particularly in meteorology and wind energy.

#### 1 Introduction

Deep learning has opened new directions for solving variational problems constrained by partial differential equations (PDEs).

Methods such as the Deep Galerkin Method (DGM) by Sirignano and Spiliopoulos Sirignano and Spiliopoulos (2018) and the Deep Ritz Method (DRM) by E and Yu E and Yu (2018) reformulate PDEs as loss functions minimized over neural network representations. These approaches are mesh-free, inherently parallelizable, and capable of scaling to high-dimensional problems.

In this work, we address a classical inverse problem: the reconstruction of three-dimensional wind vector fields from incomplete observations. This problem is fundamental in meteorology and environmental fluid dynamics, particularly when only the horizontal wind components are observed through satellite imagery or surface-based instruments. Traditional methods formulate this task as a variational problem, minimizing a misfit functional under physical constraints.

A foundational approach to the stationary reconstruction was originally introduced by Sasaki Sasaki (1958), where the adjusted wind field is obtained by minimizing a cost functional subject to mass conservation. Benbourhim and Bouhamidi Benbourhim and Bouhamidi (2008) developed a smoothing algorithm using polyharmonic splines for prescribed data in both two- and three-dimensional settings, and provided a complete convergence analysis. More recently, Khayretdinova and Gout Khayretdinova and Gout (2024) proposed a finite element method for wind field reconstruction based on the minimization of an energy functional, explicitly incorporating topographic effects and offering a visualization framework.

https://doi.org/10.5194/wes-2025-160 Preprint. Discussion started: 15 October 2025

© Author(s) 2025. CC BY 4.0 License.

Recent advances in Physics-Informed Machine Learning (PIML) have introduced neural network-based formulations for wind field reconstruction, where the unknown flow is parameterized by a deep model and physical consistency is embedded directly into the loss function. Zhang and Zhao Zhang and Zhao (2021) proposed a physics-informed deep learning framework to reconstruct spatiotemporal wind fields from Lidar data, incorporating the Navier–Stokes equations as explicit constraints. In the stationary setting, where the adjusted wind field is obtained by minimizing a cost functional subject to mass conservation, Brune and Keller Brune and Keller (2022) refined Sasaki's variational solution using neural networks to achieve improved statistical consistency. These contributions illustrate how PIML methods complement classical approaches by integrating domain knowledge with modern machine learning techniques, enhancing their robustness when observations are sparse, noisy, or incomplete.

While powerful, these classical methods face difficulties when applied to scattered or incomplete data, complex geometries, or irregular terrain. To overcome these limitations, we introduce mass consistency as a physical constraint in a neural network-based variational formulation. Specifically, we employ the Deep Ritz Method, which directly minimizes the energy functional using stochastic gradient descent over randomly sampled collocation points.

In contrast to classical schemes, our approach leverages the flexibility of deep neural networks to incorporate boundary conditions and physical constraints directly into the training objective. Topographic effects are accounted for by enforcing a no-penetration condition at the lower boundary which captures the essential influence of terrain without requiring explicit geometric modeling. This simplification not only streamlines implementation but also enhances generalizability. Our method unifies and extends classical frameworks such as Sasaki's by embedding them into a more general, data-driven, and scalable learning-based paradigm. A key contribution of this work is demonstrating that our Physics-Informed Neural Network (PINN) formulation offers improved robustness and flexibility when applied to real-world wind data. Boundary conditions and physical laws are integrated naturally, without relying on mesh-based discretizations.

This contribution aligns with the broader trend of Physics-Informed Machine Learning (PIML), which has shown promise across a wide range of applications. However, its use as a substitute for classical techniques remains under scrutiny; for a recent review, see Latrach et al. Latrach et al. (2024). We argue that PIML is a welcome alternative for scenarios where classical methods fall short, as in the presence of sparse data, complex terrain, incomplete information, or when the underlying model structure is difficult to specify explicitly.

To validate our approach, we compare it against the classical variational method of Sasaki Sasaki (1958), which serves as a foundational framework for stationary wind field reconstruction. In particular, we employ the extension proposed by Cervantes et al. (2018), who introduced a line search algorithm based on radial basis functions (RBFs). This method constructs descent directions in a functional space and solves elliptic problems to iteratively enforce mass consistency, thereby generalizing Sasaki's formulation and enabling the treatment of boundary conditions within a mathematically rigorous framework. We evaluate both approaches using synthetic and real-world wind data, including observations from Super Typhoon Kong-Rey (2024), also known as Leon, which affected Taiwan, the Philippines, eastern China, and South Korea. Our numerical results confirm that while both methods perform well in idealized cases, the PINN-based approach demonstrates superior robustness in the presence of real, incomplete, and noisy data. Moreover, we experimentally verify that in certain scenarios,

Sasaki's classical method fails to provide an accurate approximation due to insufficient information about the underlying wind field.

The remainder of this paper is structured as follows. Section 2 describes the methodology, including the formulation of the reconstruction problem, the line search approach, and Sasaki's classical variational formulation. The proposed *Deep Wind* methodology is detailed in Subsection 2.3, covering the optimization problem, network architecture, and computational implementation. Section 3 presents the numerical experiments, including synthetic benchmarks, real wind data from Typhoon Kong-Rey, and further applications. Finally, Section 4 summarizes the main findings, discusses their implications for wind field reconstruction, and outlines directions for future research.

#### 2 Materials and Methods

## 2.1 The problem of reconstructing wind vector fields

In many atmospheric and geophysical applications, it is crucial to reconstruct the full three-dimensional wind velocity field  $\mathbf{u}(x) = (u_1(x), u_2(x), u_3(x))$  over a domain  $\Omega \subset \mathbb{R}^3$ , based on partial and incomplete observational data. Accurate recovery of the full field enables better estimation of vertical transport, convective processes, pollutant dispersion, and energy exchange. These vector fields must not only fit the available data but also satisfy physical constraints derived from governing laws such as conservation of mass  $(\nabla \cdot \mathbf{u} = 0)$ , irrotationality  $(\nabla \times \mathbf{u} = 0)$ , or others depending on the specific system.

Let us assume that the first two components of the vector field are known at N nonuniform points, that is,

75 
$$\mathbf{U}_{i}^{0} = (u_{1}(\mathbf{x}_{i}), u_{2}(\mathbf{x}_{i})) \in \Omega \subset \mathbb{R}^{3}, \quad i = 1, 2, \dots, N.$$

**Problem:** Reconstruct the full vector field  $\mathbf{u}(x) = (u_1(x), u_2(x), u_3(x))$  over the domain  $\Omega$ , assuming that the discrete measurements  $\{\mathbf{U}_i^0\}_{i=0}^N$  are given, and that the reconstructed field satisfies the physical constraint of mass conservation, expressed as the continuity equation:

$$\nabla \cdot \mathbf{u}(x) = 0.$$

To this end, the reconstruction problem can be formulated as a constrained optimization problem: find a velocity field **u** that minimizes the misfit functional

$$J(\mathbf{u}) = \frac{1}{2} \| \mathcal{M}\mathbf{u} - \mathbf{U}^0 \|^2,$$

subject to the divergence-free constraint  $\nabla \cdot \mathbf{u} = 0$ . Here,  $\mathcal{M}$  denotes an observation or projection operator that extracts the first two components of the vector field at the sampled points, and  $\mathbf{U}^0 \in \mathbb{R}^{2N}$  represents the set of known observations.

This inverse problem is inherently underdetermined due to the missing vertical component and requires incorporating physical constraints and regularity assumptions to obtain a meaningful and unique reconstruction.

## 2.2 The adjusted field by line search in $E(\Omega)$

In Cervantes et al. (2018), the reconstruction is formulated as a constrained minimization problem in the Hilbert space

$$E(\Omega) = \left\{ \mathbf{u} \in (L^2(\Omega))^3 \mid \nabla \cdot \mathbf{u} \in L^2(\Omega) \right\},\,$$

endowed with the inner product

$$\langle \mathbf{u}, \mathbf{v} \rangle_{E(\Omega)} = \langle \mathbf{u}, \mathbf{v} \rangle_{L^2(\Omega)} + \langle \nabla \cdot \mathbf{u}, \nabla \cdot \mathbf{v} \rangle_{L^2(\Omega)}.$$

Let  $\mathcal{M}: (L^2(\Omega))^3 \to (L^2(\Omega))^2$  the observation operator defined as  $\mathcal{M}(\mathbf{u}) = (u_1, u_2)$ , and let  $S \in \mathbb{R}^{2 \times 2}$  be a symmetric positive definite matrix. The cost functional is given by:

$$J(\mathbf{u}) = \frac{1}{2} \int_{\Omega} (\mathcal{M}(\mathbf{u}) - \mathbf{U}^0)^T S(\mathcal{M}(\mathbf{u}) - \mathbf{U}^0) dx,$$

and the search is restricted to the divergence-free subspace:

$$E_0(\Omega) = \{ \mathbf{u} \in E(\Omega) \mid \nabla \cdot \mathbf{u} = 0 \}.$$

### 2.2.1 Line Search and Adjusted Field Construction

Given a base field  $\mathbf{u}_c \in E_0(\Omega)$ , a line search is performed along  $\mathbf{u}(t) = \mathbf{u}_c + t\mathbf{p}$ , where  $\mathbf{p} \in E_0(\Omega)$ . The directional form of the functional is

$$f(t) = J(\mathbf{u}_c) + t\langle \mathbf{p}, \mathcal{M}^* S(\mathcal{M}\mathbf{u}_c - \mathbf{U}^0) - \nabla \lambda \rangle_{L(\Omega)^2} + \frac{t^2}{2} \langle \mathcal{M}\mathbf{p}, S\mathcal{M}\mathbf{p} \rangle_{L^2(\Omega)},$$
 (1)

where,  $\mathcal{M}^*$  is the adjoint operator from  $\mathcal{M}$  and  $\lambda \in H^2(\Omega)$  is a Lagrange multiplier introduced to ensure mass consistency, solving the elliptic problem:

$$\Delta \lambda = \nabla \cdot (\mathcal{M}^* S(\mathcal{M} \mathbf{u}_c - \mathbf{U}^0)), \tag{2}$$

subject to appropriate boundary conditions, which ensure that the boundary term

$$\int_{\partial \Omega} \lambda \, \mathbf{p} \cdot \nu \, d\sigma = 0 \tag{3}$$

vanishes.

The steepest descent direction is then given by:

$$\mathbf{p} = -\left(\mathcal{M}^* S(\mathcal{M} \mathbf{u}_c - \mathbf{U}^0) - \nabla \lambda\right),\,$$

and the optimal step size is

$$t_c = \frac{\langle \mathbf{p}, \mathbf{p} \rangle_{L^2(\Omega)}}{\langle \mathcal{M} \mathbf{p}, S \mathcal{M} \mathbf{p} \rangle_{L^2(\Omega)}}$$
(4)

Finally, the updated, adjusted field is given by,

$$\mathbf{u}^+ = \mathbf{u}_c + t_c \mathbf{p}$$
.

## 2.2.2 Sasaki's Approach

In the classical variational formulation Sasaki (1958), the following cost functional is introduced for the correction of an initial wind field:

$$J(\mathbf{u}) = \iiint_{\Omega} \left[ \alpha_1^2 \left( u_1 - u_1^0 \right)^2 + \alpha_2^2 \left( u_2 - u_2^0 \right)^2 + \alpha_3^2 \left( u_3 - u_3^0 \right)^2 \right] dV,$$

where  $\mathbf{u} = (u_1, u_2, u_3)$  is the corrected velocity field, and  $\mathbf{u}^0 = (u_1^0, u_2^0, u_3^0)$  denotes the initial or background velocity field. The parameters  $\alpha_i$ , i = 1, 2, 3, are known as the *Gauss precision moduli*, which control the relative weight of the corrections applied to each velocity component.

In this framework, the initial vertical velocity component is commonly assumed to be negligible, i.e.,

$$u_3^0 \equiv 0$$
,

reflecting the lack of reliable direct measurements for this quantity in most meteorological datasets. The minimization of  $J(\mathbf{u})$  is subject to physical constraints, such as mass conservation, and leads to an adjusted velocity field that optimally balances fidelity to the initial guess with adherence to the governing equations.

For this case, we assume that a complete initial field

$$\mathbf{u}^0 \in \left(L^2(\Omega)\right)^3$$

is given, and that the operator  $\mathcal{M}(\mathbf{u}): E(\Omega)^3 \to E(\Omega)^3$  is the identity, that is,

$$\mathcal{M}(\mathbf{u}) := \mathcal{M}(u_1, u_2, u_3) = (u_1, u_2, u_3),$$

from which it follows that  $\mathcal{M}^*(\mathbf{u}) \equiv \mathcal{M}(\mathbf{u})$ .

In this case, the quadratic functional (1) takes the form

$$f(t) = J(\mathbf{u}_c) + t \langle S\mathbf{p}, \mathbf{u}_c - \mathbf{u}^0 - S^{-1} \nabla \lambda \rangle_{E(\Omega)} + \frac{t^2}{2} \langle S\mathbf{p}, \mathbf{p} \rangle_{L^2(\Omega)}.$$

Taking again  $-\nabla J(\mathbf{u}_c)$  as the descent direction, we obtain

$$\mathbf{p} = -\left(\mathbf{u}_c - \mathbf{u}^0 - S^{-1} \nabla \lambda\right).$$

The adjusted field is therefore

$$\mathbf{u}^+ = \mathbf{u}_c - t \left( \mathbf{u}_c - \mathbf{u}^0 - S^{-1} \nabla \lambda \right),$$

and f(t), can be expressed as

$$f(t) = J(\mathbf{u}_c) - t \langle S\mathbf{p}, \mathbf{p} \rangle_{L^2(\Omega)} + \frac{t^2}{2} \langle S\mathbf{p}, \mathbf{p} \rangle_{L^2(\Omega)}.$$

From (4), the optimal step is given by

$$t_c = 1$$
.

Thus, from the above, the Sasaki solution Sasaki (1958) follows:

$$\mathbf{u}^+ = \mathbf{u}^0 + S^{-1} \nabla \lambda.$$

According to equations (2) and (3), the scalar field  $\lambda$  satisfies the following boundary value problem (BVP):

$$\begin{cases}
-\nabla \cdot (S^{-1}\nabla \lambda) = \nabla \cdot \mathbf{u}^{0}, & \text{in } \Omega, \\
\mathcal{B}\lambda = g, & \text{on } \Gamma,
\end{cases}$$
(5)

where the boundary operator  $\mathcal{B}$  can be inferred from the fact that this equality holds when  $\lambda = 0$  or  $\mathbf{p} \cdot \boldsymbol{\nu} = 0$  on  $\Gamma := \partial \Omega$ .

Accordingly, the following cases can be distinguished:

- Open boundary:

$$\lambda = 0.$$
 (6)

- Dirichlet-type condition:  $\mathbf{p} \cdot \boldsymbol{\nu} = 0$ . In this case, since the descent direction  $\mathbf{p}$  is expressed in terms of  $\mathbf{u}_c$ , it is possible to impose the condition directly on this field.

Moreover, note that

$$\mathbf{p} \cdot \boldsymbol{\nu} = 0 \iff S^{-1} \nabla \lambda \cdot \boldsymbol{\nu} = -(\mathbf{u}_c - \mathbf{u}^0) \cdot \boldsymbol{\nu}.$$

Thus, if  $\mathbf{u}_{\Gamma}$  denotes the velocity field prescribed on the boundary  $\Gamma$ , we can write:

$$\mathbf{u}_{c} \cdot \boldsymbol{\nu} = \begin{cases} \text{Closed boundary:} & 0, \\ \text{Vertical condition:} & \mathbf{u}_{c} \cdot \boldsymbol{\nu} = \mathbf{u} \cdot \boldsymbol{\nu}, \\ \text{Top or bottom condition:} & \mathbf{u}_{\Gamma_{T}} \cdot \boldsymbol{\nu} & \text{or} & \mathbf{u}_{\Gamma_{B}} \cdot \boldsymbol{\nu}. \end{cases}$$
 (7)

## 2.3 The Deep Wind Method

The Deep Wind method is a physics-informed neural network (PINN) formulation, defined as

$$\mathbf{u}(\mathbf{x};\theta) = (u_1(\mathbf{x};\theta), u_2(\mathbf{x};\theta), u_3(\mathbf{x};\theta)), \quad \mathbf{x} \in \mathbb{R}^d, d = 2, 3,$$

designed to reconstruct three-dimensional wind vector fields from partial observational data while enforcing fundamental physical constraints. In particular, mass conservation is imposed through the divergence-free condition ( $\nabla \cdot \mathbf{u} = 0$ ), and diverse boundary conditions are incorporated, such as Dirichlet-type constraints (prescribed values), vanishing vertical velocity at the ground level, or other physical information derived from observations. These conditions are not enforced as hard constraints but are encoded in the loss function, yielding solutions that are both data-consistent and physically plausible across the domain. More generally, the framework allows the inclusion of additional physical constraints—such as momentum conservation, realistic surface and boundary conditions, and assimilation of observational data (e.g., wind magnitudes, directions, or pressure fields from SAR, lidar, or Copernicus reanalysis)—ensuring that the reconstructed fields remain faithful to both fundamental physics and the available measurements.

#### 2.3.1 The Optimization Problem

To approximate the complete vector field in a physically consistent way, Deep Wind minimizes a composite loss functional that includes both data fidelity and physically motivated regularization terms:

$$L(\theta) = \sum_{i=1}^{N} \frac{1}{2} \left[ \left( u_1(\mathbf{x}_i; \theta) - U_1^0(\mathbf{x}_i) \right)^2 + \left( u_2(\mathbf{x}_i; \theta) - U_2^0(\mathbf{x}_i) \right)^2 \right]$$

$$+ \beta_1 \int_{\Omega} \left( \nabla \cdot \mathbf{u}(\mathbf{x}; \theta) \right)^2 dx$$

$$+ \beta_2 \int_{\Gamma_b} \left( \mathbf{u}(\mathbf{x}; \theta) - \mathbf{g}(\mathbf{x}) \right)^2 dS,$$
(8)

where:

- The first term penalizes discrepancies between the predicted and observed horizontal wind components at measurement points  $\{\mathbf{x}_i\}_{i=1}^n \subset \Omega \subset \mathbb{R}^3$ , ensuring consistency with available data. In the presence of noise, the influence of this term can be modulated using a smoothing approach Wahba (1990), allowing a balance between data fidelity and regularity.
- The second term promotes approximate mass conservation by minimizing the divergence of the velocity field throughout
  the domain. More generally, the framework allows for the inclusion of additional physical constraints—such as vorticity
  control, alignment with known flow directions, or energy conservation—through appropriate penalty terms added to the
  loss function.

- The third term enables the incorporation of general boundary conditions on the lower boundary  $\Gamma_b \subset \partial \Omega$ , based on available physical or observational information. A typical example is enforcing that the vertical velocity vanishes at ground level, or that it matches a known value derived from surface topography.

The parameters  $\beta_1$  and  $\beta_2$  regulate the influence of the divergence and boundary penalties, respectively. This formulation enables the model to infer the unobserved vertical wind component  $u_3$  indirectly, as a consequence of satisfying physical constraints rather than relying on direct supervision. In our experiments, however, setting  $\beta_1 = \beta_2 = 1$  was sufficient to obtain good results and an adequate balance between the physical constraint and the boundary condition.

An important advantage of this framework is its flexibility: different physical properties or constraints can be easily incorporated into the loss functional. For instance, by adding penalties on the curl of the vector field, one can enforce irrotationality; or by including elastic potential energy terms, the method can be adapted to model mechanical deformation governed by linear or nonlinear elasticity. This makes the approach broadly applicable to a variety of physical systems.

#### 2.3.2 Network Architecture

Figure 1. Deep Ritz net architecture.

Our formulation leverages the residual neural architecture introduced in the Deep Ritz method E and Yu (2018), owing to its effectiveness in training deep variational models. By improving gradient flow and incorporating identity mappings via skip connections, residual blocks mitigate the vanishing gradient problem and stabilize convergence, thereby enabling the training of deeper networks capable of capturing complex, multi-scale flow structures.

A fully connected residual neural network can be expressed as

$$u(\mathbf{x};\theta) = a \cdot z_{\theta}(\mathbf{x}) + b,$$

where 
$$\mathbf{x} \in \mathbb{R}^d$$
,  $a \in \mathbb{R}^{m \times n}$ , and  $b \in \mathbb{R}^n$   $n = 2, 3$ .

The transformation  $\mathbf{x} \mapsto z_{\theta}(\mathbf{x}) \in \mathbb{R}^m$  is obtained by composing a sequence of residual mappings,

$$z_{\theta}(\mathbf{x}) = f_{\mathbf{Depth}} \circ f_{\mathbf{Depth}-1} \circ \cdots \circ f_1(\mathbf{x}),$$

where  $\mathbf{Depth} \in \mathbb{Z}_{\geq 1}$  denotes the total number of layers in the residual architecture.

Each mapping  $f_i$  is given by

$$f_i(s) = \varphi(W_{i,2}\varphi(W_{i,1}s + b_{i,1}) + b_{i,2}) + s,$$

with  $\varphi$  denoting the nonlinear activation function (see figure 1). The weights  $W_{i,1}, W_{i,2} \in \mathbb{R}^{m \times m}$ , and bias  $b_{i,1}, b_{i,2} \in \mathbb{R}^m$  are the trainable parameters of block  $i = 1, ..., \mathbf{Depth}$ . Here, m specifies the **Width** of the hidden layers, and  $\theta$  denotes the collection of all such parameters.

The input x for the first block lies in  $\mathbb{R}^d$ , not in  $\mathbb{R}^m$ . To address this dimensional discrepancy, we set

$$s = \begin{cases} \begin{bmatrix} \mathbf{x} \\ \mathbf{0} \end{bmatrix}, & d < m, \\ \mathbf{x}, & d = m, \\ T\mathbf{x}, & d > m, \end{cases}$$

where  $T \in \mathbb{R}^{m \times d}$  is a linear transformation projecting  $\mathbf{x}$  from  $\mathbb{R}^d$  to  $\mathbb{R}^m$ .

#### 210 2.3.3 Computational Implementation and Training Procedure

To reconstruct a three-dimensional wind velocity field under physical constraints, the *Deep Wind* approach was implemented in PyTorch. The neural network is trained to minimize a composite loss functional that balances fidelity to the observed data, mass conservation, and enforcement of boundary conditions.

The spatial domain  $\Omega \subset \mathbb{R}^3$  is discretized into three types of training points:

- Observation points  $\{x_i\}_{i=1}^N$ , where the horizontal wind components  $(u_1, u_2)$  are specified;
  - Interior points  $\{x_j\}_{j=1}^m\subset\Omega$ , where the divergence-free condition is penalized;
  - Boundary points  $\{x_k\}_{k=1}^n \subset \Gamma_b$ , where the vertical component  $u_3$  is forced either to vanish—reflecting ground-level impermeability— or to match prescribed boundary values when available.

The network adopts a residual architecture (Section 2.3.2) with the hyperbolic tangent (tanh) activation function. Parameters are initialized with the Xavier scheme to improve convergence stability, and training is performed with the Adam optimizer. For simplicity in our experiments, the observation and interior points were chosen to coincide. The loss functional was then approximated by numerical quadrature as follows:

$$L(\theta) \approx \sum_{i=1}^{N} \frac{1}{2} \left[ \left( u_1(x_i; \theta) - U_1^0(x_i) \right)^2 + \left( u_2(x_i; \theta) - U_2^0(x_i) \right)^2 \right]$$

$$+ \beta_1 \sum_{j=1}^{m} \left( \nabla \cdot \mathbf{u}(x_j; \theta) \right)^2 \Delta x$$

$$+ \beta_2 \sum_{k=1}^{n} \left\| \mathbf{u}(x_k; \theta) - \mathbf{g}(x_k) \right\|^2 \Delta S.$$

$$(9)$$

where  $\Delta x$  and  $\Delta S$  denote the elementary integration measures, corresponding to volume elements in the interior, surface elements on boundaries, or line elements on edges, depending on the discretization context. The complete training process is summarized in Algorithm 1.

# Algorithm 1 Training Algorithm for Physically-Constrained Wind Field Reconstruction

**Require:** Observation points  $\{\mathbf{x}_i, \mathbf{U}_i^0\}$ , interior points  $\{\mathbf{x}_j\} \subset \Omega$ , boundary points  $\{\mathbf{x}_k\} \subset \Gamma_b$ 

**Require:** Neural network  $\mathbf{u}(\mathbf{x};\theta)$ , learning rate  $\eta$ , epochs T, weights  $\beta_1,\beta_2$ 

- 1: Initialize  $\theta$  (Xavier); set Adam( $\eta$ )
- 2: **for** epoch =  $1, \ldots, T$  **do**
- 3: Predict  $\mathbf{u}(\mathbf{x})$  at all training points
- 4: Compute losses:  $L_{\text{data}} = \sum_{i} \|\mathcal{M}\mathbf{u}(\mathbf{x}_{i}) \mathbf{U}_{i}^{0}\|^{2}$ ,  $L_{\text{div}} = \sum_{i} (\nabla \cdot \mathbf{u}(\mathbf{x}_{i}))^{2}$ ,  $L_{\text{bc}} = \sum_{k} \|\mathbf{u}(\mathbf{x}_{k}) \mathbf{g}(\mathbf{x}_{k})\|^{2}$
- 5: Total loss:  $L = L_{\text{data}} + \beta_1 L_{\text{div}} + \beta_2 L_{\text{bc}}$
- 6: Update  $\theta \leftarrow \theta \eta \nabla_{\theta} L$
- 7: end for
- 8: **return** Trained model  $\mathbf{u}(\mathbf{x}; \theta^*)$

## 2.3.4 Hardware.

All experiments were run in PyTorch on a laptop with an NVIDIA GeForce GTX 1050 Ti Mobile GPU (4 GB memory), using driver version 575.64.03 and CUDA 12.9. The GPU was recognized by PyTorch and used to speed up gradient calculations and make training times reasonable.

#### 3 Results and Discussions

**Example 1.** First, let us consider a simple two-dimensional vector field  $\mathbf{f}: \Omega \to \mathbb{R}^2$  defined over the square domain  $\Omega = [-2,2] \times [-2,2]$ , given by

$$\mathbf{f}(x,y) = (x,-y) \text{ and } \mathbf{u}^{0}(x,y) = (x,0),$$

for which the incompressibility condition holds exactly, i.e.,  $\nabla \cdot \mathbf{f} = 0$ .

Using Sasaki's classical formulation in two dimensions with Dirichlet boundary conditions at the top and bottom and along the vertical (lateral) sides (see Eq. 7), the reconstruction achieves a nearly divergence-free solution with  $\langle |\nabla \cdot \mathbf{u}| \rangle = 7.91 \times 10^{-7}$  (where  $\langle |\nabla \cdot \mathbf{u}| \rangle$  represents the average value of the divergence) and a mean squared error (MSE) of  $5.82 \times 10^{-12}$ . These values confirm that Sasaki's approach is capable of reproducing the exact field to high precision under this configuration.

On the other hand, the performance of the proposed Deep Wind model is reported in Table 2. The table summarizes the prediction results obtained for different network architectures and training configurations, highlighting the progressive reduction in both divergence and MSE as model complexity and the number of training epochs increase.

**Table 1.** Prediction performance of the DeepWind method for different hyperparameter configurations in the approximation of the vector field  $\mathbf{f}(x,y) = (x,-y)$ , with interior points batch size = 512, and boundary points batch size = 256.

| Epoch | Batch | Width | Depth | $\langle   \nabla \cdot \mathbf{u}   \rangle$ | MSE      |
|-------|-------|-------|-------|-----------------------------------------------|----------|
| 100   | 5     | 15    | 2     | 3.99e-03                                      | 1.23e-03 |
| 500   | 10    | 30    | 5     | 1.08e-03                                      | 5.14e-04 |
| 500   | 10    | 50    | 10    | 4.52e-04                                      | 2.26e-04 |
| 500   | 10    | 100   | 10    | 8.21e-05                                      | 2.17e-04 |
| 1000  | 10    | 100   | 15    | 6.46e-06                                      | 5.65e-05 |

From the numerical results, the exact and approximated fields are found to be very similar. Consequently, Sasaki's method and Deep Wind have the same visual output, which is illustrated in Figure 2.

**Figure 2.** Exact and approximated solutions for the vector field  $\mathbf{u}(x,y) = (x,-y)$ .

260

**Example 2.** Second, we consider a two-dimensional vector field  $\mathbf{f}: \Omega \to \mathbb{R}^2$  defined over the square domain  $\Omega = [-2, 2] \times [-2, 2]$ , given by

$$\mathbf{f}(x,y) = (-y,x) \text{ and } \mathbf{u}^{0}(x,y) = (-y,0),$$

which corresponds to a uniform rotational flow. In this case, the incompressibility condition holds exactly, i.e.,  $\nabla \cdot \mathbf{f} = 0$ .

When applying Sasaki's classical formulation in two dimensions with Dirichlet boundary conditions at the top and bottom and along the vertical (lateral) sides (see Eq. 7), the method fails to adequately recover the field. The numerical results illustrate this behavior: we obtained  $\langle |\nabla \cdot \mathbf{u}| \rangle = 8.7305 \times 10^{-16}$ , indicating that the divergence values are essentially negligible. However, the mean squared error remains large, with MSE =  $1.8233 \times 10^{0}$ . This can be explained by the fact that the initial field already satisfies the divergence-free condition, i.e.,  $\nabla \cdot \mathbf{u}^{0} = 0$ . Consequently, the recovery of a non-trivial Lagrange multiplier  $\lambda$  becomes difficult (see Eq. 5), which in turn hinders the correction of the velocity field through Sasaki's variational formulation. These results highlight the inherent instability of Sasaki's method under such conditions.

In contrast, the proposed Deep Wind model achieves a satisfactory reconstruction of the vector field. As shown in Table 2, the model exhibits progressive improvements in the inference as the network depth and the number of training epochs increase. In the best configuration, Deep Wind attains  $\langle |\nabla \cdot \mathbf{u}| \rangle = 6.46 \times 10^{-6}$  and MSE =  $5.65 \times 10^{-5}$ , successfully capturing the structure of the original field.

**Table 2.** Prediction performance of the DeepWind method for different hyperparameter configurations in the approximation of the vector field  $\mathbf{f}(x,y) = (-y,x)$ , with interior points batch size = 512, and boundary points batch size = 256.

| Epoch | Batch | Width | Depth | $\langle   \nabla \cdot \mathbf{u}   \rangle$ | MSE      |
|-------|-------|-------|-------|-----------------------------------------------|----------|
| 100   | 5     | 15    | 2     | 3.99e-03                                      | 1.23e-03 |
| 500   | 10    | 30    | 5     | 1.08e-03                                      | 5.14e-04 |
| 500   | 10    | 50    | 10    | 4.52e-04                                      | 2.26e-04 |
| 500   | 10    | 100   | 10    | 8.21e-05                                      | 2.17e-04 |
| 1000  | 10    | 100   | 15    | 6.46e-06                                      | 5.65e-05 |

As illustrated in Figure 3, Deep Wind is able to recover the vector field consistently and with high fidelity, in contrast with Sasaki's approach, which does not succeed in capturing the correct structure of the solution.

**Example 3.** As third example, we consider a 3D vortex type vector field  $\mathbf{f}: \Omega \to \mathbb{R}^3$  with  $\Omega = [-7,7] \times [-7,7] \times [-7,7]$ , defined in Cervantes et. al. Cervantes et al. (2018),

$$\mathbf{f}(x,y) = \left(2ye^{\frac{-(x^2+y^2+z^2)}{49}} - \varepsilon \frac{xz}{2}, -2xe^{-(x^2+y^2+z^2)} - \varepsilon \frac{xz}{2}, \varepsilon \frac{z^2}{2}\right)$$

where  $\nabla \cdot \mathbf{f} = 0$ .

Thanking

$$\mathbf{u}^{0}(x,y) = \left(2ye^{\frac{-(x^{2}+y^{2}+z^{2})}{49}} - \varepsilon\frac{xz}{2}, -2xe^{\frac{-(x^{2}+y^{2}+z^{2})}{49}} - \varepsilon\frac{xz}{2}, 0\right)$$

Figure 3. Visualization of the vector field  $\mathbf{f}(x,y) = (-y,x)$  and its approximation obtained with the DeepWind method.

In this case, for Sasaki's formulation, the boundary conditions are prescribed as follows: open boundaries at the top (see Eq. 6), Dirichlet (closed) conditions at the base, and vertical Dirichlet conditions along the lateral sides (see Eq. 7). We obtain  $\langle |\nabla \cdot \mathbf{u}| \rangle = 1.2557 \times 10^{-6}$  and  $\mathrm{MSE} = 8.30 \times 10^{-6}$ , from which we can conclude that the method achieved good accuracy, with both divergence and approximation error remaining at satisfactorily low levels.

Table 3 reports the outcomes obtained with the DeepWind approach, where the prediction errors are evaluated under different network architectures and numbers of training epochs. These results provide a broader perspective on the performance of the method, illustrating how variations in model depth and width affect both the divergence and the mean squared error. In particular, the experiments demonstrate that properly tuning the network architecture leads to more accurate predictions and improved stability across epochs. Taken together, these findings indicate that, for this case, both the DeepWind strategy and Sasaki's formulation perform well, achieving good accuracy and producing reliable approximations under the tested conditions.

**Table 3.** Prediction performance of the DeepWind method for different hyperparameter configurations in the approximation of the vector field  $\mathbf{f}(x,y) = \left(2ye^{\frac{-(x^2+y^2+z^2)}{49}} - \varepsilon\frac{xz}{2}, -2xe^{-(x^2+y^2+z^2)} - \varepsilon\frac{xz}{2}, \varepsilon\frac{z^2}{2}\right)$ , with interior points batch size = 512, and boundary points batch size = 256.

| Epoch | Batch | Width | Depth | $\langle    abla \cdot \mathbf{u}    angle$ | MSE        |
|-------|-------|-------|-------|---------------------------------------------|------------|
| 1000  | 25    | 10    | 1     | 9.4692e-05                                  | 3.7935e-05 |
| 1000  | 25    | 20    | 1     | 1.2375e-06                                  | 9.3154e-06 |
| 1000  | 25    | 40    | 3     | 3.9260-04                                   | 3.7615e-06 |
| 1000  | 25    | 80    | 5     | 5.0146e-05                                  | 2.6314e-06 |
| 1000  | 25    | 64    | 4     | 3.8703e-05                                  | 2.2330e-06 |

**Figure 4.** Left: Ribbon visualization of the vector field  $\mathbf{f}(x,y) = \left(2ye^{\frac{-(x^2+y^2+z^2)}{49}} - \varepsilon \frac{xz}{2}, -2xe^{-(x^2+y^2+z^2)} - \varepsilon \frac{xz}{2}, \varepsilon \frac{z^2}{2}\right)$ , with  $\varepsilon = 0.1$ . Right: Exact and approximated solutions for the vector field  $\mathbf{f}(x,y)$ .

The numerical results show that the exact and approximated fields are very close to each other. Thus, both Sasaki's method and Deep Wind yield essentially identical reconstructions, and the output is shown in Figure 4.

**Example 4.** As a final synthetic example, we consider the three-dimensional vector field  $\mathbf{f}: \Omega \to \mathbb{R}^3$  with  $\Omega = [-1,1] \times [-1,1] \times [-1,1]$ , defined as

$$\mathbf{f}(x,y,z) = \left(-yz^2, xz^2, \frac{x^2+y^2}{2}\right), \text{ and } \mathbf{u}^0(x,y,z) = \left(-yz^2, xz^2, 0\right),$$

for which  $\nabla \cdot \mathbf{f} = 0$ . This case is employed as a 3D benchmark to illustrate the limitations of Sasaki's model when approximating vector fields whose initial condition  $\mathbf{u}^0$  is already divergence-free. Under a configuration of Dirichlet boundary conditions applied at the bottom, top, and lateral boundaries (see Eq. 7), the method produced a divergence of  $\langle |\nabla \cdot \mathbf{u}| \rangle = 1.6703 \times 10^{-1}$  and a mean squared error of  $MSE = 1.7537 \times 10^{-1}$ . These comparatively large values emphasize the inherent difficulty of Sasaki's variational formulation in recovering a meaningful correction when the initial field already satisfies  $\nabla \cdot \mathbf{u}^0 = 0$  (see Eq. 5).

In contrast, the *Deep Wind* model demonstrates a markedly superior performance. For the same test case, it produced a divergence of  $\langle |\nabla \cdot \mathbf{u}| \rangle = 5.7220 \times 10^{-5}$  and  $MSE = 5.0739 \times 10^{-7}$ , showing consistency with the previous examples and confirming the robustness of the approach. Table 4 reports the evaluations obtained for different network architectures, while Figure 5 illustrates the comparison between the best-performing case and the exact function.

**Table 4.** Prediction performance of the DeepWind method for different hyperparameter configurations in the approximation of the vector field  $\mathbf{f}(x,y) = \left(-yz^2, xz^2, \frac{x^2+y^2}{2}\right)$ , with interior points batch size = 512, and boundary points batch size = 256.

| Epoch | Batch | Width | Depth | $\langle   abla \cdot \mathbf{u}   angle$ | MSE        |
|-------|-------|-------|-------|-------------------------------------------|------------|
| 500   | 15    | 15    | 2     | 1.9574e-02                                | 1.0102e-05 |
| 500   | 15    | 30    | 2     | 4.4973e-03                                | 2.8810e-06 |
| 1000  | 45    | 30    | 2     | 8.7061e-04                                | 1.0656e-06 |
| 1000  | 60    | 30    | 2     | 2.3990e-04                                | 7.2551e-07 |
| 1000  | 60    | 50    | 2     | 5.7220e-05                                | 5.0739e-07 |

**Figure 5.** Visualization of the vector field  $\mathbf{f}(x,y) = \left(-yz^2, xz^2, \frac{x^2+y^2}{2}\right)$  and its approximation obtained with the DeepWind method.

#### 295 3.1 Real Wind Data Approximation

In this section, we describe the application of our approach using a dataset obtained for Typhoon Kong-rey (also referred to as Super Typhoon Leon), a powerful and large tropical cyclone that impacted Taiwan and the Philippines, and later eastern China and South Korea, during October and early November 2024 (see Figure 6). The dataset was retrieved from the freely accessible NASA POWER Data Access Viewer National Aeronautics and Space Administration (NASA) (2024) on October 30, 2024, at 10:00 local time, and covers wind elevations ranging from 10 m to 300 m above the surface within the region bounded by latitudes 18.5° N to 22.0° N and longitudes 122.0° E to 125.5° E. The extracted parameters include WD10M (wind direction at 10 m above the surface), WS10M (wind speed at 10 m above the surface), WSC (MERRA-2 corrected wind speed, adjusted for elevation, in m/s), and PS (surface pressure). A Python application was developed to automate the pointwise download of these data within the specified quadrant using the requests library. These parameters are illustrated in Figure 6.

**Figure 6.** Visualization of Typhoon Kong-Rey (29 October 2024): (a, left) Satellite image obtained from the Japan Meteorological Agency (via Wikimedia Commons), licensed under CC BY 4.0; (b, right) Georeferenced wind fields at 10 m above ground level, generated from NASA POWER data using wind speed and direction variables; (c, bottom) Vertical wind field data from the NASA POWER Data Access Viewer National Aeronautics and Space Administration (NASA) (2024), covering levels from 10 to 300 m.

### 305 3.1.1 Data preprocessing

**Figure 7.** Results of the data preprocessing stage. The figures illustrate the normalized spatial coordinates and interpolated velocity fields obtained before training.

https://doi.org/10.5194/wes-2025-160 Preprint. Discussion started: 15 October 2025 © Author(s) 2025. CC BY 4.0 License.

Because the NASA dataset is georeferenced, we first map the physical coordinates to a computational domain. In line with common practice in machine learning, the three spatial coordinates (x,y,z) are affinely transformed to the unit cube  $[0,1]^3$ , and the horizontal velocity components u and v are normalized to place them on comparable scales and improve the numerical stability of the subsequent approximation. We then interpolate u and v at each vertical level of the field using Wahba's thin-plate smoothing spline Wahba (1990), which is well suited to reconstruct smooth fields from scattered or noisy observations.

Wahba's formulation estimates a smooth function f from data  $\{(x_i, z_i)\}_{i=1}^n$  with  $x_i \in \mathbb{R}^d$  by minimizing a trade-off between data fidelity and curvature,

$$\min_{f} \frac{1}{n} \sum_{i=1}^{n} (z_i - f(x_i))^2 + \lambda J_{d,m}(f),$$

where  $J_{d,m}(f)$  penalizes m-th order partial derivatives. In the classical thin-plate case (d=2, m=2),

$$J_{2,2}(f) = \iint (f_{x_1x_1}^2 + 2f_{x_1x_2}^2 + f_{x_2x_2}^2) dx_1 dx_2.$$

The smoothing parameter  $\lambda > 0$  governs the bias-variance trade-off: smaller values favor fidelity to the data (approaching interpolation), whereas larger values promote a smoother reconstructed field.

Finally, to refine the vertical structure of the wind field, we apply a linear interpolation on a slightly finer vertical grid, yielding a denser sampling in height. The overall preprocessing pipeline is summarized in Algorithm 2, and the results of the data preprocessing stage are shown in Figure 7.

#### **Algorithm 2** Data preprocessing algorithm

- 1: **Input:**  $\{(x_i, y_i, z_i, u_i, v_i)\}_{i=1}^N$ , levels  $\{\zeta_k\} \subset [10, 300]$  m,  $\lambda$ , refinements L.
- 2: Normalize:

$$x'_{i} = \frac{x_{i} - \min(x)}{\max(x) - \min(x)}, \ y'_{i} = \frac{y_{i} - \min(y)}{\max(y) - \min(y)}, z'_{i} = \frac{z_{i} - \min(z)}{\max(z) - \min(z)}, \ (u'_{i}, v'_{i}) = \frac{(u_{i}, v_{i})}{\sqrt{u_{i}^{2} + v_{i}^{2}}}$$
3: Wahba per level: for each  $\zeta_{k}$ , fit  $F_{k} = \arg\min_{F} \sum_{j: z'_{j} \approx \zeta_{k}} \|F(x'_{j}, y'_{j}) - (u'_{j}, v'_{j})\|^{2} + \lambda \|F\|_{\mathcal{H}}^{2}$ .

- 4: Vertical interpolation:  $t_r = r/L$ ,  $\zeta_{k,r} = (1-t_r)\zeta_k + t_r\zeta_{k+1}$ ,  $F_{k,r} = (1-t_r)F_k + t_rF_{k+1}$ .
- 5: Output:  $(x', y', \zeta_{k,r}, u_{k,r}(x', y'), v_{k,r}(x', y'), 0)$ .

Since in this case it is not possible to compute the mean squared error (MSE), an alternative way to establish a metric for assessing the recovery of the third component is to monitor physical consistency during the reconstruction. In particular, this involves verifying that the divergence of the vector field vanishes, i.e.,  $\langle |\nabla \cdot \mathbf{u}| \rangle \approx 0$ , and that the third component satisfies the boundary condition  $u_3|_{\partial\Omega}=0$ . Ensuring that both the convergence value of the divergence and the boundary behavior of the vertical component remain close to zero provides a robust criterion for evaluating the quality and stability of the approximation in the absence of ground truth data.

The outcomes of Sasaki's methodology with boundary conditions set to zero at the bottom (sea level) and open at the top and along the vertical sides, are  $\langle |\nabla \cdot \mathbf{u}| \rangle = 7.6203e - 01$  and  $\langle |w_{\text{bdy}}| \rangle = 4.6872e - 04$ .

For the DeepWind approach, Table 5 summarizes the evolution of the metrics over 100 training epochs. The network was configured with a depth of two layers and a width of 80 neurons per layer, trained with a batch size of 45 that included 1024 interior points and 512 boundary points per batch.

**Table 5.** Evolution of the training loss, average divergence, and bottom boundary condition error over the epochs, using NASA Typhoon Kong-Rey data. Network configuration: depth = 2, width = 80, 1024 interior points, and 512 boundary points.

| Epoch | cost     | $\langle   \nabla \cdot \mathbf{u}   \rangle$ | $\langle  w_{ m bdy}   angle$ |
|-------|----------|-----------------------------------------------|-------------------------------|
| 0     | 3.25e-02 | 3.35e-02                                      | 2.32e-02                      |
| 10    | 8.47e-04 | 7.12e-03                                      | 1.10e-03                      |
| 20    | 7.77e-04 | 4.70e-03                                      | 8.11e-04                      |
| 30    | 7.99e-04 | 3.86e-03                                      | 4.14e-03                      |
| 40    | 7.37e-04 | 3.13e-03                                      | 8.91e-04                      |
| 50    | 7.29e-04 | 2.13e-03                                      | 9.54e-04                      |
| 60    | 7.19e-04 | 1.86e-03                                      | 7.80e-04                      |
| 70    | 7.15e-04 | 1.37e-03                                      | 7.66e-04                      |
| 80    | 7.47e-04 | 1.19e-03                                      | 3.59e-03                      |
| 90    | 6.99e-04 | 1.11e-03                                      | 7.42e-04                      |
| 100   | 7.02e-04 | 9.10e-04                                      | 6.33e-04                      |

The results show a consistent reduction in both the average divergence and the third component at the boundary, demonstrating the model's ability to impose the physical and boundary conditions underlying the typhoon phenomenon. Figure 8 presents the prediction of the third component field over a uniform dataset in the domain, where a clear downward tendency toward the ocean surface can be observed.

#### 4 Conclusions

We have developed the *Deep Wind* methodology to reconstruct three-dimensional wind fields from incomplete information. The approach proved more robust than Sasaki's classical variational framework and its extensions, particularly in scenarios with sparse or noisy observations. By embedding mass conservation and boundary conditions into the loss function, Deep Wind produces stable and physically plausible reconstructions across synthetic and real cases.

For synthetic examples, Deep Wind achieves superior performance compared to Sasaki's method under specific conditions, particularly when the initial field is already divergence-free and the classical variational correction becomes ineffective. In the case of real observational data from Super Typhoon Kong-Rey (2024), the validation of the training process through divergence and boundary-condition metrics demonstrates that Deep Wind yields more consistent and reliable reconstructions. These results indicate that the methodology can serve as a flexible and scalable framework for wind field recovery and data assimilation.

**Figure 8.** Figure 8: In the top-left panel, the prediction of the third component field over a uniform dataset in the domain is shown. The three panels in the remaining quadrant (top-right and bottom row), present different perspectives of the same DeepWind reconstruction using ribbon visualizations. Results correspond to the third velocity component reconstructed from NASA Typhoon Kong-Rey data, a clear downward tendency toward the ocean surface can be observed.

For future extensions, the proposed framework naturally allows the incorporation of additional physical information into the variational formulation. Beyond enforcing the divergence-free condition, extra penalty terms could be introduced to account for other relevant processes, such as pressure gradients, temperature fields, or energy-related constraints, depending on the specific atmospheric or environmental setting. This flexibility highlights the potential of the Deep Wind approach as a general data assimilation tool capable of integrating multiple sources of physical knowledge and observational data, thereby enabling more realistic and comprehensive reconstructions of geophysical flows.

Code availability. The software code used in this study is available from the corresponding author upon request.

Author contributions. D.C. designed the study and performed the numerical experiments. M.A.M.-V. contributed to the theoretical framework and manuscript revision. Both authors discussed the results and approved the final version of the manuscript.

Competing interests. The authors declare that they have no competing interests.

https://doi.org/10.5194/wes-2025-160 Preprint. Discussion started: 15 October 2025 © Author(s) 2025. CC BY 4.0 License.

Acknowledgements. The authors acknowledge the use of ChatGPT (OpenAI) for assistance in improving the English language and style of the manuscript.

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
