# Peer review of "The Deep Wind Method: Physics-Informed Wind Field Reconstruction with Mass Consistency"

_Wind Energy Science, 2025_

## Referee Comment (RC1)

The paper proposes Deep Wind, a physics-informed neural network to reconstruct 3-D wind fields from incomplete observations while embedding mass consistency and boundary conditions. The method is compared against some benchmarks on several synthetic cases and one real case based on a typhoon dataset. Below are my major comments:

- 1. The horizontal components are normalized by their local magnitude, i.e.,  $(u',v')=(u,v)/\sqrt{u^2+v^2}$ . This appears to remove speed information and retain only direction, How the original speed magnitudes are recovered for evaluation/visualization?
- 2. The typhoon dataset includes levels "from 10 m to 300 m," but the extracted parameters are WD10M, WS10M, WSC, PS. As written, this is a little ambiguous. Does the model use the higher-level winds or whether vertical profiles are obtained via interpolation only?
- 3. The penalties parameters are set  $\beta_1=\beta_2=1$ . Could you please justify this fixed choice? It would be better to include a sensitivity analysis (e.g.,  $\beta_1,\beta_2\in\{0.1,1,10,100\}$ ).
- 4. Please define all operators and symbols. While readers familiar with PINNs may infer them, others in data science, statistics, or general machine learning may not.
- 5. In Figs. 2–5, I would suggest using thinner arrows (and consistent color scales) to make small errors more visible.